# Accurate Robot Arm Attitude Estimation Based on Multi-View Images and Super-Resolution Keypoint Detection Networks

**DOI:** 10.3390/s24010305

**Published:** 2024-01-04

**Authors:** Ling Zhou, Ruilin Wang, Liyan Zhang

**Affiliations:** College of Mechanical & Electrical Engineering, Nanjing University of Aeronautics and Astronautics, Nanjing 210016, China; zhouling_ttyy@nuaa.edu.cn (L.Z.); wangrl97@foxmail.com (R.W.)

**Keywords:** robot arm, attitude estimation, super-resolution keypoint detection network (SRKDNet), multi-view images

## Abstract

Robot arm monitoring is often required in intelligent industrial scenarios. A two-stage method for robot arm attitude estimation based on multi-view images is proposed. In the first stage, a super-resolution keypoint detection network (SRKDNet) is proposed. The SRKDNet incorporates a subpixel convolution module in the backbone neural network, which can output high-resolution heatmaps for keypoint detection without significantly increasing the computational resource consumption. Efficient virtual and real sampling and SRKDNet training methods are put forward. The SRKDNet is trained with generated virtual data and fine-tuned with real sample data. This method decreases the time and manpower consumed in collecting data in real scenarios and achieves a better generalization effect on real data. A coarse-to-fine dual-SRKDNet detection mechanism is proposed and verified. Full-view and close-up dual SRKDNets are executed to first detect the keypoints and then refine the results. The keypoint detection accuracy, PCK@0.15, for the real robot arm reaches up to 96.07%. In the second stage, an equation system, involving the camera imaging model, the robot arm kinematic model and keypoints with different confidence values, is established to solve the unknown rotation angles of the joints. The proposed confidence-based keypoint screening scheme makes full use of the information redundancy of multi-view images to ensure attitude estimation accuracy. Experiments on a real UR10 robot arm under three views demonstrate that the average estimation error of the joint angles is 0.53 degrees, which is superior to that achieved with the comparison methods.

## 1. Introduction

In the context of intelligent manufacturing, robot arms with multiple joints play increasingly important roles in various industrial fields [1,2]. For instance, robot arms are utilized to accomplish automatic drilling, riveting and milling tasks in aerospace manufacturing; in automobile and traditional machinery manufacturing fields, robot arms can be frequently seen in automatic loading/unloading, automatic measurement and other production or assembly tasks. 

In most industrial applications, a robot arm works in accordance with the pre-planned program. However, on occasions where the robot arm becomes out of control by mistake, serious collision or injury accidents may occur, especially in the work context of human–machine cooperation. Therefore, it is critical to configure monitoring means to ensure safety. On-site attitude monitoring of working robot arms is also essential for the collaborative work of multiple robot arms.

Machine vision is one of the most suitable and widely used monitoring means due to its relatively low cost, high applicability and good accuracy. To reduce the difficulty of image feature recognition and to improve monitoring accuracy and reliability, a common method in industry is to arrange cooperative visual targets on the monitored object [3,4]. However, arranging cooperative targets is usually cumbersome and time-consuming, and the targets in industrial sites tend to suffer from being stained or falling off. Studying methods for accurately estimating the attitude of robot arms without relying on cooperative visual markers presents a significant research challenge [5,6].

With their excellent ability to extract image features, deep neural networks have been widely used in the field of computer vision. They can extract natural feature information and deep semantic information from images and realize various computer vision tasks based on the rich extracted information without relying on cooperative visual markers. The base of the robot arm in most industrial scenes is fixed on the ground or a workbench. In this situation, the motion attitude of the robot arm is completely determined by the rotation angle of each joint. Therefore, monitoring the robot arm attitude is essential to determine the rotation angles of the arm joints. One approach to this is constructing an end-to-end neural network model to directly predict the attitude parameters of the robot arm through the input image of the robot arm. However, the end-to-end method requires more computing resources. In addition, it is not easy to make full use of the kinematic constraints of the robot arm and the imaging constraints from 2D to 3D space. Therefore, the attitude estimation accuracy of the end-to-end method is difficult to ensure. Another possible approach to attitude estimation is composed of two stages. First, the feature points of the robot arm are detected in the image, and then system equations are established to solve the angle of each joint. This strategy can better leverage the advantages of deep learning and 3D machine vision theory.

Keypoint detection is a major application of deep learning methods. Toshev et al. [7] directly located the image coordinates of the keypoints on the human body through convolutional neural networks to determine the human pose. Instead of outputting determined positions for the detected keypoints, the subsequent keypoint detection networks commonly output the positions in the form of heatmaps [8,9,10,11]. Newell et al. [8] proposed SHNet (Stacked Hourglass Network), which stacked several hourglass modules and detected keypoints based on multi-scale image feature information. Chen et al. [9] proposed CPNet (Cascaded Pyramid Network), which is cascaded by two convolutional neural network modules. The first module is used to detect all keypoints, and the second module corrects the poor-quality results detected by the first module to improve the final detection accuracy. Peng et al. [10] proposed PVNet (Pixel-wise Voting Network), which can obtain superior keypoint detection results when the target object is partially blocked. Sun et al. [11] proposed HRNet (High-Resolution Network), which processes feature maps under multiple-resolution branches in parallel, so that the network can maintain relatively high-resolution representation in forward propagation. These neural networks have found successful applications in tasks such as human body posture estimation, where qualitative understanding instead of quantitative accuracy is the main concern. Due to the huge demand for computing resources for network training, these neural networks have to resort to a downsampling process for front-end feature extraction, which leads to the resolution of the output heatmaps being insufficient for high-accuracy estimation.

Determining how to make the neural network output a higher-resolution heatmap without significantly increasing the consumption of computing resources is a significant problem worth investigating. Super-resolution image recovery based on deep learning has seen great progress in recent years. SRCNNet (Super-Resolution Convolutional Neural Network) [12], VDSRNet (Very Deep Super-Resolution Network) [13], EDSRNet (Enhanced Deep Super-Resolution Network) [14], etc., have been proposed. The early super-resolution reconstruction networks need to upsample a low-resolution input image to the target resolution for subsequent processing before training and prediction; therefore, the computational complexity is high. Shi et al. [15] proposed ESPCNNet (Efficient Subpixel Convolutional Neural Network) from the perspective of reducing computational complexity. This convolutional neural network deals with low-resolution feature maps in the training process and only adds a subpixel convolution layer to realize upsampling operation in the end, which effectively increases the speed of super-resolution image reconstruction. It has the potential to improve the resolution of the output keypoint heatmaps and in turn improve the keypoint positioning accuracy by introducing the idea of super-resolution image reconstruction into the keypoint detection network.

To monitor the attitude of a robot arm, it is essential to solve the rotation angle of each joint. Based on the depth image of the robot arm, Widmaier et al. [16] used a random forest regression model to estimate the robot arm attitude. Labbe et al. [17] and Zuo et al. [18] estimated the robot arm attitude based on one single grayscale image. However, serious joint occlusion is inevitable in one single-perspective image, which makes it hard to detect some keypoints and may even lead to wrong estimation results. Moreover, the depth ambiguity problem in monocular vision may lead to multiple solutions in attitude estimation, reducing the monitoring reliability of the robot arm.

In this paper, we present a two-stage high-precision attitude estimation method for base-fixed six-joint robot arms based on multi-view images. The contributions include the following: (1) A new super-resolution keypoint detection network (SRKDNet for short) is proposed. The novelty of the SRKDNet lies in that a subpixel convolution module is incorporated in the backbone neural network HRNet [11] to learn the law of resolution recovery of the downsampled feature maps. This method can alleviate the disadvantages of low-resolution heatmaps and improve the keypoint detection accuracy without significantly increasing the computing resource consumption. (2) A coarse-to-fine detection mechanism based on dual SRKDNets is put forward. A full-view SRKDNet obtains a relatively rough keypoint detection result. Then, a close-up SRKDNet is executed to refine the results with a cropped image of the ROI determined by the results of the full-view SRKDNet. The dual-SRKDNet detection mechanism performs better than one-time detection, and the keypoint detection accuracy is drastically improved. (3) Efficient virtual-and-real sampling and neural network training methods are proposed and verified. The virtual sample data are first used to train the neural network, and then a small number of real data are applied to fine-tune the model. This method achieves accurate keypoint detection for real data without consuming a huge amount of time and manpower. (4) The constraint equations for solving the rotation angles of each joint are established; they depict the relation among the detected keypoints in the multi-view images, the camera imaging model and the kinematic model of the robot arm. A screening strategy based on the keypoint detection confidence is incorporated in the solving process and is proved to be critical for ensuring attitude estimation accuracy. Experiments demonstrate that the whole set of methods proposed in this paper can realize high-accuracy estimation of robotic arm attitude without utilizing cooperative visual markers.

The remaining contents of this paper are arranged as follows: In Section 2, we introduce the whole set of methods, including the approaches to high-precision keypoint detection (Section 2.1), automatic virtual sample generation (Section 2.2) and robot arm attitude estimation (Section 2.3). Experiments on virtual and real robot arms are reported in Section 3. We conclude the paper in Section 4.

## 2. Materials and Methods

### 2.1. High-Precision Detection of Keypoints

The first step of the two-stage attitude estimation method proposed in this paper is to detect the preset keypoints on/in the robot arm in the images. The detection accuracy directly affects the accuracy of the joint angle estimation.

The preset keypoints were selected under three basic criteria: (1) The keypoints should have distinctive features to be identified in the images. (2) The keypoints should be helpful for determining the attitude of the robot arm. (3) There are at least 2 keypoints on each rod. Taking the commonly used UR10 robot arm as the research object, we selected 20 keypoints on/in the robot arm (including the working unit attached at the end). The keypoint set was composed of the 3D center points (the red points) of each rotating joint, the 3D midpoint (the blue points) of each rod segment and some salient feature points (the green points) of the arm, as shown in Figure 1. The first two types of points are inside the structure, and the third type of points is on the surface of the structure. The keypoints form a skeleton, which can effectively characterize the attitude of the entire robot arm. 

Each keypoint was directly selected on the 3D digital model of the robot arm when it was in the zero position. To generate the sample data for neural network training, either a virtual or real robot arm was controlled to move to the specified positions. The 3D coordinates of the preset keypoints at each specified position (for instance, the position as in Figure 1) could be obtained according to the kinematics of robotic arms [19], which will be detailed in Section 2.2.2. 

Given any predefined keypoint position in/on the digital model, its corresponding image point can be calculated according to the camera model. In this way, we obtained a large number of training samples for the keypoint detection network. Experiments show that with the predefined keypoints, the keypoint detection network works well and the arm attitude estimation achieves high accuracy.

This section begins with a brief introduction to the backbone network HRNet [11] used in this paper. Then, we introduce the idea of image super-resolution reconstruction into keypoint detection and propose a super-resolution keypoint detection network, SRKDNet, which can alleviate the disadvantages of the low-resolution heatmaps without significantly increasing the computing resource consumption. A coarse-to-fine keypoint detection scheme based on dual SRKDNets is also presented in this section.

#### 2.1.1. Brief Introduction to HRNet

To retain high resolution for the feature maps in the forward propagation, the high-resolution network HRNet processes feature maps at various resolutions, as shown in Figure 2. First, a preprocessing module is used to downsample the input image, which lowers the resolution of the output heatmaps as well. The main structure of HRNet can be divided into several stages, and the branches at different resolutions in each stage use the residual blocks to extract features. After each stage, a new branch is created without abandoning the original-resolution branch. The new branch is obtained by strided convolutions. The length and width of the new feature map are reduced to 1/2 of the original, but the number of channels becomes twice that of the original. In the new stage, the feature maps are created by fusing the multi-scale feature maps of each branch in the previous stage. The HRNet shown in Figure 2 has four branches with different resolutions. The final output feature map integrates the information extracted from the four branches and is used for generating the keypoint heatmap. 

The multi-scale fusion operation in HRNet is shown in Figure 3. The feature maps from the branches with the same resolution remain the same while those with different resolutions are converted to the same resolution first via upsampling or downsampling (strided convolutions). Then, they are aggregated to obtain the output maps. HRNet has powerful multi-scale image feature extraction capability and has been widely used in classification recognition, semantic segmentation and object detection. We take HRNet as the backbone in our super-resolution keypoint detection network (SRKDNet) which will be presented in the next subsection.

#### 2.1.2. Super-Resolution Keypoint Detection Network (SRKDNet)

If the resolution of the sample images is large, the neural networks will consume huge amounts of computing resources in the training. Therefore, HRNet and most related neural networks like SHNet [8] utilize a preprocessing convolution module to downsample the input images. Taking HRNet for instance, the original images need to be downsampled to 64 × 64 through the preprocessing convolution module. The so-called high-resolution feature map maintained in each subsequent layer of the convolutional neural network structure is only 64 × 64. If an ordinary upsampling method, e.g., neighbor interpolation, is used to restore the resolution, the downsampling operation will lead to precision loss of the keypoint positions in the heatmaps. Suppose the coordinates of a keypoint in the original image are (u,v), and the downsampling scale is a; then, the coordinates of the keypoint on the corresponding ground-truth heatmap are (u/a,v/a), which have to be rounded as ([u/a],[v/a]). Here, [·] represents the round operator. When the keypoint coordinates ([u/a],[v/a]) are re-mapped back to the original resolution denoted as $\left(u^{\prime },v^{\prime }\right)$, a maximum error is possibly generated:(1)$$\varepsilon_{\max}=\sqrt{{\left(u-u^{\prime }\right)}^{2}+{\left(v-v^{\prime }\right)}^{2}}=\frac{\sqrt{2}}{2}a$$

It can be seen that the larger the downsampling scale is, the more possible accuracy loss of the keypoint position in the heatmap there will be. 

Inspired by the methods of image super-resolution reconstruction, we propose a super-resolution keypoint detection network (SRKDNet). As shown in Figure 4, SRKDNet uses HRNet as the backbone network to extract multi-scale feature information in the images. The preprocessing module extracts shallow features from the input image and downsamples it. The generated low-resolution feature map is then sent to the core module of HRNet. Instead of directly sending the feature maps output by the core module to the 1 × 1 convolution layer for heatmap generation, SRKDNet incorporates a subpixel convolution layer [15] after the core module to learn the law of resolution recovery of the downsampled feature maps. In addition, a branch is added to the input side, and the resolution of the output feature map of the branch is the same as that of the input image (also called the original resolution). The branch is combined with the resolution-recovered feature map output by the resolution recovery module through channel fusion processing. Finally, a 1 × 1 convolutional layer is used for generating the heatmap with the original resolution.

Adding a branch at the input side can provide additional shallow image features for the final heatmap generation. We believe that the combination of shallow and deep feature information is conducive to keypoint detection. In order to avoid the loss of image information, the branch consists of only one convolution layer and one batch normalization layer, with no activation layer involved.

The essence of subpixel convolution is to rearrange the pixels from a low-resolution image with more channels in a specific way to form a high-resolution image with fewer channels. As shown in Figure 5, pixels of different channels at the same position on the low-resolution image with size $\left(r^{2},H,W\right)$ are extracted and composed into small squares with size (r,r), which together form a high-resolution image with size (1,rH,rW) Realizing the subpixel convolution requires convoluting the low-resolution feature maps first to expand the number of image channels. For instance, the low-resolution image in size of (C,H,W) needs to be expanded to size $\left(r^{2}C,H,W\right)$ via convolution before it can be converted to a high-resolution image in size of (C,rH,rW). In our implementation, r=2. 

Since the backbone network HRNet has a strong ability to detect features in multiple scales, SRKDNet does not adopt a complex super-resolution neural network structure. Instead, only a subpixel convolution module is applied to enable the convolutional neural network to learn the knowledge for generating high-quality and high-resolution heatmaps from low-resolution information. The subsequent experiments can prove its significant effects in improving the neural network performance.

#### 2.1.3. Coarse-to-Fine Detection Based on Dual SRKDNets

For collecting the sample data of the robot arm, it is necessary to make the field of view of the camera completely cover the working space of the robot arm, which is much larger than the robot arm itself. Therefore, the robot arm occupies only a small region in some images, while the large background regions have no help in the keypoint detection. To further improve the detection performance, we propose a coarse-to-fine detection strategy based on dual SRKDNets. First, an SRKDNet, namely a full-view SRKDNet, is trained by using the original sample images, as shown in Figure 6a. By using the trained full-view SRKDNet, the coarse keypoint detection results are obtained (red points in Figure 6b). Based on the relatively rough detection results, the corresponding region of interest (ROI) of the robot arm in each image (blue bounding box) is determined, as shown in Figure 6b. According to the ROI, a new image is cropped from the sample image, as shown in Figure 6c. Using the cropped new images as the sample data, another SRKDNet, namely close-up SRKDNet, is trained. The two SRKDNets use the same convolutional neural network structure, the same training flow and the same setup. For an image to be detected, the trained full-view SRKDNet is first used for rough detection. The cropped image of the robot arm ROI is then put into the trained close-up SRKDNet to obtain the final keypoint detection result, shown as the blue points in Figure 6d. Our experiments have demonstrated that the detection scheme based on dual SRKDNets can drastically improve the keypoint detection performance. The details of the experiments can be found in Section 3.

### 2.2. Automatic Sample Generation Based on Virtual Platform

#### 2.2.1. Virtual Platform Construction

The training of a neural network requires a large number of sample data. The predictive effect of the neural model is directly related to the quantity and quality of the sample data. To make the neural network fully “know” the robot arm to be detected in the image and make the trained model have a more stable performance, the sample images should be taken from various perspectives, under various backgrounds and lighting conditions. Obviously, collecting a large number of diverse sample data in real industrial scenarios will consume a lot of manpower and time.

For this paper, a virtual platform in UE4 [20] was established to simulate the working scene of the UR10 robot arm equipped with a working unit. The color, roughness, high brightness and metallicity of the appearance of the real robot arm are presented in the platform as far as possible. The base coordinate system of the UR10 robot arm is set as the world coordinate system. The robot arm in the zero position is shown in Figure 7a. A movable skeleton and a parent–child relationship between the adjacent bones are created according to the structure and kinetic characteristics of the robot arm, as shown in Figure 7b. Each bone has a head and a tail node. The head node is connected to the parent bone, and the tail node is connected to the child bone. Each bone can be rotated freely with the head node as a reference point, and the child bone connected to its tail node moves with it together. The 3D digital model of each arm segment is bound to the corresponding bone, so as to drive the articulated arm to move together with the skeleton, as shown in Figure 7c.

In UE4, the motion posture and speed of the robot arm can be easily set; one or more virtual cameras can be deployed in the scene; the internal and external parameters of the cameras, as well as the lighting conditions and the background of the scene, can be flexibly changed. In this way, we virtually collected a large number of sample data under various backgrounds and lighting conditions for training the SRKDNets. The background settings are randomly selected from the images in the COCO dataset [21]. A moving light source is used in the constructed virtual platform. The position and properties of the light source keep changing during the sample data collection. Rich background settings and lighting conditions in the sample data can make the neural network insensitive to the background/lighting changes and more focused on extracting the features of the robot arm itself. 

After the virtual scene with the robot arm is established, the virtual cameras take virtual images of the scene to obtain the synthetic image of the robot arm. These virtual images will serve as the training samples. The attitude parameters of the robot arm, as well as the internal and external parameters of the virtual cameras corresponding to each virtual image, are recorded for image labeling, which will be detailed in the next subsection. Figure 8 shows three typical virtual sample images of the robot arm.

#### 2.2.2. Automated Generation and Labeling of Virtual Samples

With the virtual platform constructed in Section 2.2.1, we generated a large number of virtual sample images using a Python program, utilizing the UnrealCV library [22] and the blueprint script in UE4. The collection process of a single sample is as follows:The rotation angles of each joint of the robot arm, the pose parameters of the virtual camera, the light intensity and direction, and the background are randomly generated. They are used to automatically update the virtual sample collecting scene in UE4.A virtual image of the current virtual scene is taken via the virtual camera.For any keypoint $P_{j}$ on/in the *m*-th arm segment, its relative coordinate Pjm in the arm segment coordinate system $C_{m}$ is converted to Pj0 in the robot base coordinate system $C_{0}$ according to the successive parent–child transformation relationship of the bone segments: (2)P˜j0=Tm0(θ1,θ2,⋯,θm)P˜jm
where P˜j0 is the homogeneous form of Pj0, P˜jm is the homogeneous form of Pjm and Tm0(θ1,θ2,θm)∈R4×4 is the transformation matrix from $C_{m}$ to $C_{0}$, which is determined by the rotation angles $\theta_{1},\theta_{2},\theta_{m}$ of the m joints. The coordinate values of Pjm do not change with the movement of the robot arm and can be determined in advance according to the digital model of the robot arm.According to the internal and external parameters of the virtual camera, the pixel coordinates of each keypoint on the virtual image are calculated by using the camera imaging model in Formula (3)
(3)sp˜j=K[R  t]P˜j0
where ${\tilde{p}}_{j}$ represents the homogeneous pixel coordinate vector of keypoint $P_{j}$ in the virtual image, P˜j0 is the homogeneous form of the 3D coordinates in the robot base coordinate system $C_{0}$, R and t are the rotation matrix and translation vector from $C_{0}$ to the virtual camera coordinate system $C_{c}$, K is the intrinsic parameter matrix of the camera and s is a scalar coefficient.According to the pixel coordinates $p_{j}$ of each keypoint $P_{j}$, the heatmap label of the current virtual image is generated. In the generated heatmap of $P_{j}$, the weight is set to the largest value at $p_{j}$ and gradually decreases around $p_{j}$ with Gaussian distribution. All the heatmaps of $P_{j}(j=1,2,\ldots \ldots ,J)$ are concatenated as the training label of the current virtual sample.

In addition, data enhancements, including random image rotation, translation, scaling and gray changes, are also carried out on the generated sample images to improve the robustness and generalization of the model.

Using the above process, we automatically collected a large number of virtual sample data with training labels.

### 2.3. Attitude Estimation Based on Multi-View Images

In view of the inevitability of the keypoint occlusions and the lack of depth constraints in a single-view image, we used multi-view images combined with the proposed keypoint learning algorithm to estimate the attitude of the robot arm and verify its performance on accuracy and reliability.

#### 2.3.1. Solving Rotation Angles of Robot Arm Joints

As stated in Section 2.2.2, for the keypoint $P_{j}$ on the *m*-th arm segments, its 3D coordinate Pj0 in the base coordinate system $C_{0}$ under any robot attitude can be calculated from the 3D coordinate Pjm in $C_{m}$ according to Equation (2). Suppose that L cameras are arranged to monitor the robot arm from different perspectives, then by combining Equations (2) and (3), we have the following:(4)sjp˜jl=KlRl  tlTm0(θ1,θ2,⋯,θm)P˜jm,  j=1,2,⋯,J;l=1,2,⋯,L
where ${\tilde{p}}_{j}^{l}$ denotes the homogeneous pixel coordinates of the keypoint $P_{j}$ in the *l*-th camera’s image plane; $K^{l},\;R^{l},\;t^{l}$ represent the intrinsic and extrinsic parameters of the *l*-th camera; and $\theta_{m}$ is the rotation angle of the *m*-th joint. For all the keypoints in the multi-view images, Formula (4) forms an equation system composed of L×J equations.

In the robot arm attitude monitoring process, the image coordinates $p_{j}^{l}$ (l=1,2,⋯,L) of the keypoints $P_{j}$(j=1,2,⋯,J) in the L images are located via the proposed dual SRKDNets; the camera parameters $K^{l},\;R^{l},\;t^{l}$ are known in advance (In the virtual experiments, the camera parameters can be directly obtained from the settings. In the real experiments, the intrinsic parameters are determined with the popular calibration method presented in [23]. The relative geometry relationship between the UR10 robot arm and the cameras was calibrated in advance with the well-established off-line calibration method presented in [24,25].); the 3D coordinates of Pjm are determined on the 3D digital model of the robot arm. Therefore, after removing the scale factor $s_{j}$, the unknowns in the equation system (4) are only the rotation angles of the joints. The LM (Levenberg–Marquardt) algorithm [26] can be used to optimize the equation system to obtain the joint angles $\theta_{1},\theta_{2},\cdots ,\theta_{m}$. The initial values of $\theta_{1},\theta_{2},\cdots ,\theta_{m}$ are randomly assigned within their effective ranges.

#### 2.3.2. Keypoint Screening Based on Detection Confidence

Some keypoints, especially those on the first segment or on the flange of the robot arm, are prone to be occluded by other arm segments in certain perspectives, as shown in Figure 9. When a keypoint is blocked in the image, the detection reliability of the neural network will decline, and the error between the predicted position and the real position will be larger (see Section 3 for the experimental results). The accuracy decline of the keypoint detection will inevitably increase the attitude estimation error.

However, in the case of monitoring with multi-view images, a keypoint is not likely to be occluded in all the images. Therefore, we propose a keypoint screening scheme, which is based on the detection confidence of the keypoint, to improve the attitude estimation accuracy.

As mentioned above, the value of each pixel in the heatmap output by the SRKDNet represents the probability that the image of the keypoint is located on that pixel. The pixel with the largest probability value (i.e., the detection confidence) in the heatmap will be selected as the detection result. For the L images from different perspectives, each keypoint will have L detection results, whose confidence values are different. The L detection results are sorted from high to low according to their confidence values. Then, the results with low confidence scores are discarded, and at least two results with the highest scores are kept. The screened results with high detection quality are substituted into Formula (4) so that the attitude of the robot arm can be solved more accurately and reliably. It should be noted that with this screening scheme, the number of equations in (4) will be less than L×J, but still far more than the number of unknowns. Therefore, it can ensure robust solutions. 

## 3. Experiments

### 3.1. Experiments on Virtual Data

The virtual sample acquisition and labeling methods described in Section 2.2 were used to generate 11,000 labeled sample images with a resolution of 640 × 640. We randomly selected 9000 virtual samples as the training set and 1000 virtual samples as the validation set. The validation set was not included in the training and was only used to verify the effect of the model after each round of training. The other 1000 virtual samples served as the test set to demonstrate the final effect of the model after all rounds of training. All the sample images in this study were monochrome. Before the sample images were put into the convolutional neural network for training, they were reduced to the resolution of 256 × 256. All the experiments in this study were performed on a Dell workstation with an RTX2080S graphics card and 8 GB video memory.

#### 3.1.1. Loss Function and Model Training Settings

Denote the labels in a sample batch as $\left\{G_{i}(i=1,2,\cdots ,N)\right\}$ and the heatmap output by the convolutional neural network as $\left\{H_{i}(i=1,2,\cdots ,N)\right\}$, where N represents the sample number in a batch during the training process. The number of channels, height and width of the true heatmap corresponding to a sample image are the same as those of the predicted heatmap, denoted as C, H and W, respectively. The number of channels C equals the keypoint number, i.e., 20, in this paper. 

The mean square error was used as the loss function:(5)$$MSE=\frac{1}{N}\frac{1}{C}\frac{1}{H}\frac{1}{W}\sum_{i=0}^{N-1}\sum_{j=0}^{C-1}\sum_{h=0}^{H-1}\sum_{w=0}^{W-1}{\left[G_{i}(j,h,w)-H_{i}(j,h,w)\right]}^{2}$$

SHNet [8], HRNet [11] and the proposed SRKDNet were trained with the generated virtual data for the comparison of the keypoint detection performance among these models. The PyTorch library was used to build and train the models. In the training of HRNet and the proposed SRKDNet, the settings in Ref. [11] were adopted: Adam optimizer was used; the initial learning rate was set to 0.001; the total training epoch was 45; the data batch size was 8; the learning rate was reduced once every 15 rounds with a reduction factor of 0.1. The weights of HRNet were obtained from the pre-trained HRNet on the ImageNet [27] dataset. For the backbone network of the proposed SRKDNet, the same initial weights and number of intermediate layers as in Ref. [11] were adopted. For SHNet, two hourglass modules were stacked, and its training followed the settings in Ref. [8]: the Rmsprop optimizer was used, the learning rate was initially set to 0.00025 and the neural network was trained from scratch using Pytorch’s default weight initialization.

The resolution of the heatmaps output by both SHNet and HRNet was 64 × 64. The standard deviation of the Gaussian distribution of the weights on the corresponding ground-truth heatmap was set to 1 pixel. The resolution of the heatmaps output by SRKDNet was 256 × 256, and the standard deviation of the Gaussian distribution of the weights on the corresponding ground-truth heatmap was set to 3 pixels. The channel number of the highest-resolution branch of HRNet and SRKDNet was 32. The channel number of the feature maps in the supplementary branch containing shallow image features in SRKDNet was set to 8.

#### 3.1.2. Experimental Results on Full-View SRKDNet

A commonly used metric, namely the percentage of correct keypoints (PCK) [18], was adopted to evaluate the keypoint prediction accuracy of each model. It is defined as shown in Formula (6):(6)$$PCK=\frac{1}{A}\sum_{i=1}^{A}\delta (\frac{e_{i}}{e_{norm}}),\;\delta (x)=\left\{\begin{array}{c} 1,\;x\leq \tau \\ 0,\;x>\tau \end{array}\right.$$
where A is the total number of predicted results; $e_{i}$ is the pixel distance between the predicted and the ground-truth positions; $e_{norm}$ is the standard error distance; τ is a specified threshold to adjust the ratio between the calculated error distance in the experiments and $e_{norm}$. If the calculated distance error $e_{i}$ between the predicted and the true positions of the keypoint is less than $e_{norm}\times \tau \;$, δ equals 1 and the predicted position is considered correct. The keypoint prediction result of our full-view SRKDNet will be compared with that of SHNet and HRNet by using PCK as the metric. In our experiment, $e_{norm}$ was set to 40 pixels and τ was assigned as 0.2, 0.15 or 0.1. Considering that the three neural networks output heatmaps with different resolutions, but the detected keypoint positions need to be mapped back to the original sample images to conduct the subsequent robot arm attitude estimation, we mapped the predicted coordinates of all keypoints to the original resolution 640 × 640 for comparison. Table 1 lists the PCK values of the three methods, where PCK@0.2, PCK@0.15 and PCK@0.1 represent the prediction accuracy with τ=0.2, τ=0.15 and τ=0.1, respectively. 

The results in Table 1 show that the trained full-view SRKDNet completely outperforms the two comparison models SHNet and HRNet under all three threshold values. The smaller the threshold is, the more obvious the superiority of the full-view SRKDNet over the two comparison models is. The reasons for the superiority may lie in two aspects: (1) Using the heatmaps with a higher resolution (256 × 256) in the training labels can reduce the negative influence of the downsampling operation. (2) The predictive heatmap with the trained super-resolution layer can express the detected keypoints more accurately.

Figure 10 shows the detection results of the three keypoint detection networks SHNet, HRNet and our full-view SRKDNet for the same test image. The green dots are the real locations of the keypoints, and the blue dots are the predicted locations. The mean error refers to the average of the pixel distances between the predicted locations and the real locations of all the keypoints. The mean prediction error of full-view SRKDNet is significantly lower than that of SHNet and HRNet. We can also intuitively see that most keypoint locations predicted by the full-view SRKDNet are closer to the real location than those predicted by the comparison methods.

#### 3.1.3. Experiment on Occlusion Effect

A large number of experiments have shown that the detection confidence of the occluded keypoints in the image is low. In the example shown in Figure 11, keypoint No. 1 on the end working unit is completely blocked by other arm segments. It turns out that the maximum confidence of this keypoint in the heatmap is only 0.116. Meanwhile, keypoint No. 2 on the first arm segment is visible in the image, and its detection confidence is 0.774, which is much higher than that of keypoint No. 1. It can also be clearly observed from Figure 11 that the predicted position of keypoint No. 1 deviates a lot from the real position, while the predicted position of keypoint No. 2 is closely coincident with the real position. The verified negative influence of self-occlusion on keypoint detection motivated the screening scheme proposed in Section 2.3.2. Experiments on the screening scheme will be reported in Section 3.1.5.

The GPU (graphics processing unit) memory occupation of the full-view SRKDNet was also compared with that of SHNet and HRNet with the batch size set to 8 in the training. The result is shown in Table 2. The output heatmap resolution of the three convolutional neural networks is shown in parentheses.

Table 2 shows that HRNet occupies the least GPU memory during the training and outputs heatmaps with a resolution of only 64 × 64. The SRKDNet occupies 22.6% more GPU memory resources than HRNet. This demonstrates that the proposed full-view SRKDNet can remarkably improve the detection accuracy (see Table 1) at the expense of a mild increase in GPU occupation.

For further comparison, we canceled the downsampling operations in the preprocessing stage of HRNet so that HRNet can also output heatmaps with a 256 × 256 resolution, which is the same as that of the full-view SRKDNet. However, the maximum batch size of HRNet can only be set to 2 in this situation, and the experimental results are shown in Table 3. These results demonstrate that to enable HRNet to output heatmaps with the same resolution as that of the full-view SRKDNet, the GPU memory resource consumption will increase sharply. When batch size = 2, the GPU occupation of HRNet exceeds 277.03% compared with our full-view SRKDNet.

#### 3.1.4. Experimental Results on Dual SRDKNets

In this section, we present the results of keypoint detection conducted with dual SRDKNets to verify the effect of the coarse-to-fine detection strategy. Based on the detection results of the full-view SRKDNet in Section 3.1.2 for each virtual sample image, the corresponding region of interest (ROI, i.e., the region within the bounding box of the detected keypoints) of the robot arm in each image was determined. The close-up SRKDNet was then trained using the clipped local ROI images. 

In the experiments on dual SRDKNets, for any virtual test samples, the trained full-view SRKDNet was first used to conduct initial keypoint detection. Then, the ROI image was input to the trained close-up SRKDNet to achieve the final detection results. The threshold τ was set to 0.15, 0.1 and 0.05, which were more stringent in order to adapt to the detection accuracy increase. The other settings were consistent with the experiments in Section 3.1.2. 

The keypoint detection results of the full-view SRKDNet and dual SRDKNets are shown in Table 4. In the “full-view SRKDNet” method, only the trained full-view SRKDNet was utilized for the keypoint detection. In the “dual SRKDNet” method, the trained close-up SRKDNet was used following the full-view SRKDNet.

Table 4 shows that the PCK score of dual SRKDNets is higher than that of the full-view SRKDNet, which means that the use of the close-up SRKDNet can effectively improve the keypoint detection accuracy. When a more stringent threshold is set, a more obvious improvement can be achieved. When τ = 0.05, in other words, when the distance threshold between the detected and the real keypoint positions was set to 2 pixels, the keypoint detection accuracy increased from 62.14% to 93.92%. When the threshold was assigned as 0.1, the PCK score of the close-up SRKDNet increased to 97.66%, compared to 89.07% of the full-view SRKDNet. 

The above experimental results demonstrate that the proposed successive working mechanism of the dual SRKDNets is quite effective. The close-up SRKDNet can further improve the keypoint detection accuracy by a large margin.

#### 3.1.5. Robot Arm Attitude Estimation Experiments

We used the dual SRKDNets and virtual multi-view images to verify the effect of robot arm attitude estimation. Specifically, four cameras were arranged in the UE4 virtual environment, and 1000 sets of four-perspective sample images were collected using the method in Section 2.2. The setting values of the six joint rotation angles corresponding to each of the 1000 sets of images were recorded as the ground-truth values of the 1000 attitude estimation experiments. The full-view SRKDNet trained as described in Section 3.1.2 and the close-up SRKDNet trained as described in Section 3.1.4 were used for the coarse-to-fine keypoint detection.

The comparison experiments of single-view and multi-view attitude estimation, as well as the comparison of using and not using the confidence-based keypoint screening scheme, were conducted. The specific keypoint screening method for the four-perspective sample images adopted in the attitude estimation experiments was as follows: The detection keypoints with the top three highest confidence values were kept. If the fourth detection result had a confidence score greater than 0.9, it would also be retained; otherwise, it would be discarded.

The average error of the estimated rotation angles of the 1000 experiments of each joint is shown in Table 5. “Single view” means the attitude estimation was performed based on the information from one single-perspective image (we randomly selected the 1000 sample images collected by the second camera); “four views” means that image from all four perspectives was used in the attitude estimation; “four views + confidence screening” means the multi-view keypoint screening scheme was utilized on the basis of “four-perspective”.

The above experimental results demonstrate that the average estimation error of the joint angles using images from four perspectives was reduced by 76.60% compared with that using the information from one perspective only. The confidence-based keypoint screening scheme further reduced the average error of the four-view attitude estimation by 46.23%. The compound accuracy increase reaches nearly an order of magnitude, which proves that the whole set of methods proposed in this paper is very effective.

### 3.2. Experiments on Real Robot Arm

#### 3.2.1. Real Data Acquisition

The scene of a real robot arm attitude estimation experiment is shown in Figure 12, in which three cameras are distributed around a UR10 robot arm. The intrinsic parameters of the cameras and the transformation matrix of each camera coordinate system relative to the base coordinate system of the robot arm were calibrated in advance by using well-studied methods [23,24,25]. 

We planned 648 positions for the flange endpoint of the robot arm in its working space as the sample positions, as shown in Figure 13. Each sample position corresponded to a set of six joint angles. After the robot arm reached each sample position, the three cameras collected images synchronously and automatically recorded the current joint angles and the 3D coordinates of the center endpoint of the flange in the base coordinate system. This process was repeated until the real sample data collection was completed. A total of 1944 images were captured by the three real cameras.

The resolution of the industrial camera used in the experiment was 5120 × 5120. To facilitate the training and prediction of the keypoint detection networks, and to unify the experimental standards, the resolution of the collected real images was reduced to 640 × 640, which was the same as the resolution of the virtually synthesized images. The detected keypoint positions were mapped back to the initial images for the robot arm attitude estimation.

#### 3.2.2. Keypoint Detection Experiment on Real Robot Arm

The real UR10 robot arm is consistent with the digital model in the virtual sampling platform. Therefore, all the settings in the experiments in Section 3.1, the geometric parameters of the robot arm, the 3D coordinates of the keypoints in the arm segment coordinate system and the kinematic model of the robot arm were also applied to the real robot arm attitude estimation experiments.

When the full-view SRKDNet trained using the virtual samples as described in Section 3.1.2 was used to detect the keypoints in the real images, its detection accuracy on real images was only 34.11%, 20.86% and 7.06% when the threshold τ was set to 0.15, 0.1 and 0.05, respectively. Therefore, we considered using the real sample data to fine-tune the trained model. 

From the 1944 images (648 sets of triple-view real data) obtained in Section 3.2.1, 99 real sample data (33 sets) were randomly selected as training sets. Another randomly selected 99 real sample data (33 sets) served as the validation sets. The remaining 1746 samples (582 sets) were used as the test sets to evaluate the performance of keypoint detection and attitude estimation.

The full-view SRKDNet and the close-up SRKDNet pre-trained with virtual sample data were both fine-tuned with the training sets. For comparison, we also tried the method in which the full-view SRKDNet and the close-up SRKDNet were trained not with virtual sample data but directly with the real training sets. Since the number of real samples used for training was very small, the training epoch was set to 300. The learning rate was reduced once for every 100 epochs with a reduction coefficient of 0.1. The other settings were consistent with those in Section 3.1.1. The keypoint detection results of the 1746 real test samples with the model trained with these methods are shown in Table 6. The comparison of all the experimental results was still evaluated at a resolution of 640 × 640. The first row displays the result of applying only the full-view SRKDNet trained with the 99 real sample data (training sets) to detect the keypoints. The second row displays the result of applying only the full-view SRKDNet trained with our virtual datasets and fine-tuned with the 99 real sample data (training sets) to detect the keypoints. The keypoint detection results of dual SRKDNets trained with the 99 real sample data (training sets) are shown in the third row. The last row displays the result of dual SRKDNets trained with our virtual datasets and fine-tuned with the 99 real sample data (training sets). The initial weights of the backbone network of these models were obtained from HRNet pre-trained with the ImageNet dataset.

Table 6 shows that no matter whether full-view SRKDNet only or dual SRKDNets were used for the keypoint detection, the model trained with the virtual samples generated in Section 3.1.2 and fine-tuned with 99 real sample data (training sets) demonstrates much better detection accuracy on the real test. When the full-view SRKDNet and close-up SRKDNet were trained with no virtual data but only the 99 real sample data (training sets), the keypoint detection accuracy values on the real test images were obviously lower. The results of this experiment verify the following: (1) the virtual samples generated with the proposed method in Section 2.2 have a significant positive effect on the keypoint detection of the robot arm in real scenes; (2) small amounts of real samples can efficiently re-train the model having been trained with virtual samples and achieve high generalization on real robot arms.

An example of keypoint detection in a realistic scenario is shown in Figure 14, where the first row shows the situation of using the full-view SRKDNet only, and the second row shows the situation of using the full-view SRKDNet and the close-up SRKDNet. The first column in Figure 14 shows the input real images, the second column shows the achieved heatmaps and the third column illustrates the keypoint detection results.

#### 3.2.3. Attitude Estimation Experiment on Real Robot Arm

In this section, we report the real robot arm attitude estimation experiment, which was carried out based on the keypoint detection method with the highest detection accuracy in Section 3.2.2, i.e., the last method in Table 6. The confidence-based keypoint screening scheme was employed in the triple-view attitude estimation process. For each keypoint, the results with the first and the second highest confidence scores were retained. If the confidence score of the lowest results was larger than 0.9, it was also retained; otherwise, it was discarded. The distortion compensation was conducted according to the calibrated camera parameters to rectify the detected keypoint position. 

The average estimation errors of each joint angle on the 582 real test sets collected from three perspectives are shown in Table 7. The ground-truth values of each joint angle were obtained from the control center of the robot arm when the sample data were collected. The error of each joint in the table is the average of 582 estimated results.

The experimental results in Table 7 show that the whole set of methods proposed in this paper can achieve high-precision attitude estimation for the robot arm under realistic scenarios. The total average error of all of the six joints is only 0.53°, which is even slightly better than the average error of 0.57° in the attitude estimation for the virtual test samples in Table 5. Here, we briefly analyze the reasons. When sampling in the virtual environment, each joint angle and the camera poses were set at random. Therefore, a certain number of virtual samples are unrealistic, such as interference existing between the arm segments, and positions of the arm segments being too tight, which results in self-occlusion of some keypoints in all perspectives. It is hard to estimate the attitude of the robot arm with these virtual samples. However, in the real scenario, only the normal working attitudes appear in the sample data, with no extremely strange attitudes for the sake of safety. This may explain the reason why the attitude estimation accuracy using three-view information in real scenes is slightly better than that using four-view information in a virtual dataset.

The average joint angle estimation errors of a four-joint robot arm reported in Ref. [17] and Ref. [18] are 4.81 degrees and 5.49 degrees, respectively, while the average error of the estimated joint angle of the six-joint UR10 robot arm with our method is only 0.53 degrees, which is significantly lower. The reasons may lie in three aspects: (1) The SRKDNet proposed in this paper learns the heatmaps with higher resolution by adding a subpixel convolutional layer. In addition, the combination detection scheme based on the full-view and close-up dual SRKDNets significantly improves the detection accuracy of the keypoints. (2) The existing methods only use images from one single view, while our method uses multi-view images, which can effectively alleviate the problems of self-occlusion and depth ambiguity exhibited in single-view images. Moreover, the negative influence of improper keypoint detection results can be greatly reduced by using the information redundancy of the multi-view images and the confidence-based keypoint screening scheme. (3) The existing methods not only estimate the joint angles of the robot arm but also estimate the position and attitude of the robot arm relative to the camera, so there are 10 unknowns to be estimated. Considering that the base of the robot arm is usually fixed in the industrial scenes, we determine the relative geometry relationship between the robot arm and the cameras through a well-established off-line calibration process to simplify the problem to six unknowns. The above aspects of our method together contribute to the significant accuracy improvements.

We also counted the time consumed in the two stages of keypoint detection and attitude estimation in the real scene, as shown in Table 8. The total time used for keypoint detection in the three-perspective images using the dual SRKDNets was 0.28 s. The resolution of the sample images used for the keypoint detection here was still 640 × 640. The time required for solving the joint angles of the real robot arm was 0.09 s.

## 4. Conclusions

We have proposed a set of methods for accurately estimating the robot arm attitude based on multi-view images. By incorporating a subpixel convolution layer into the backbone neural network, we put forward the SRKDNet to output high-resolution heatmaps without significantly increasing the computational resource consumption. A virtual sample generation platform and a keypoint detection mechanism based on dual SRKDNets were proposed to improve the keypoint detection accuracy. The keypoint prediction accuracy for the real robot arm is up to 96.07% for PCK@0.15 (i.e., the position deviation between the predicted and the real keypoints is within 6 pixels). An equation system, involving the camera imaging model, the robot arm kinematic model, and the keypoints detected with confidence values, was established and solved to finally obtain the rotation angles of the joints. The confidence-based keypoint screening scheme makes full use of the information redundancy of the multi-view images and is proven to be effective in ensuring attitude estimation. Plenty of experiments on virtual and real robot arm samples were conducted, and the results show that the proposed method can significantly improve the robot arm attitude estimation accuracy. The average estimation error of the joint angles of the real six-joint UR10 robot arm under three views is as low as 0.53 degrees, which is much higher than that of the comparison methods. The entire proposed method is more suitable for industrial applications with high precision requirements for robot arm attitude estimation.

In the real triple-view monitoring scenario, a total of 0.37 s was required for the keypoint detection stage and the attitude-solving stage. The keypoint detection accounted for the most time. The reason lies in that our method needs to detect keypoints in multi-view images with dual SRKDNets. Therefore, the efficiency of the proposed method is lower than that of the single-view-based method.

In this study, we only conducted experiments on one U10 robot arm. In the future, we will try to extend our method to real industrial scenes with more types of robot arms. 

## Figures and Tables

**Figure 1 sensors-24-00305-f001:**
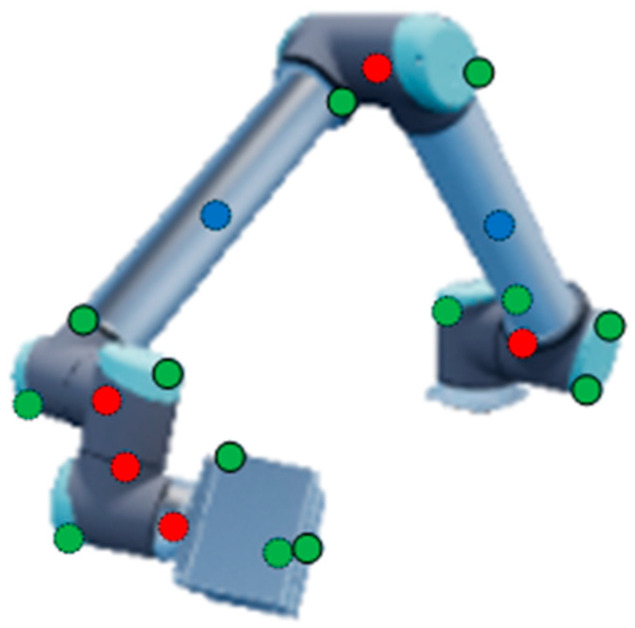
The selected keypoints on/in the robot arm.

**Figure 2 sensors-24-00305-f002:**
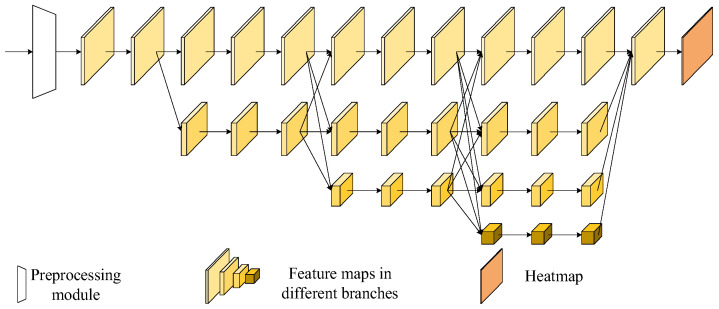
Network structure of HRNet.

**Figure 3 sensors-24-00305-f003:**
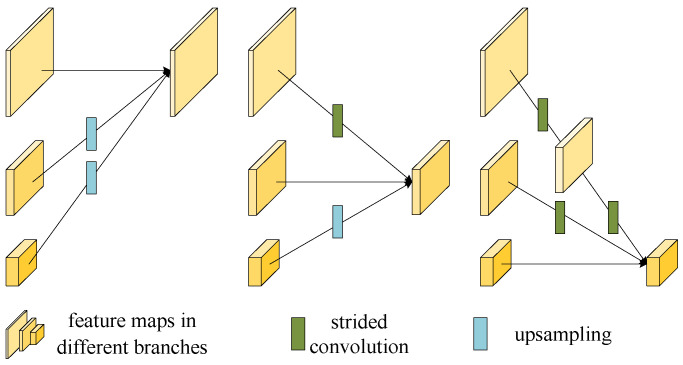
Multi-scale fusion method of HRNet.

**Figure 4 sensors-24-00305-f004:**
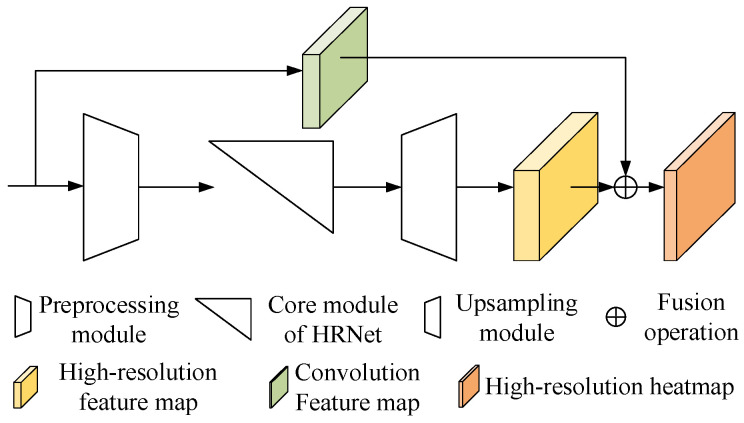
Structure of the proposed SRKDNet.

**Figure 5 sensors-24-00305-f005:**
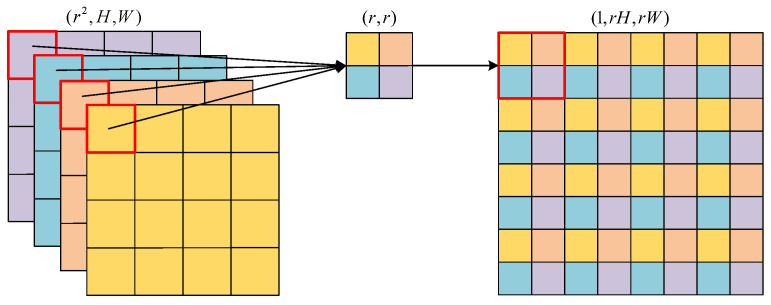
Subpixel convolution processing.

**Figure 6 sensors-24-00305-f006:**
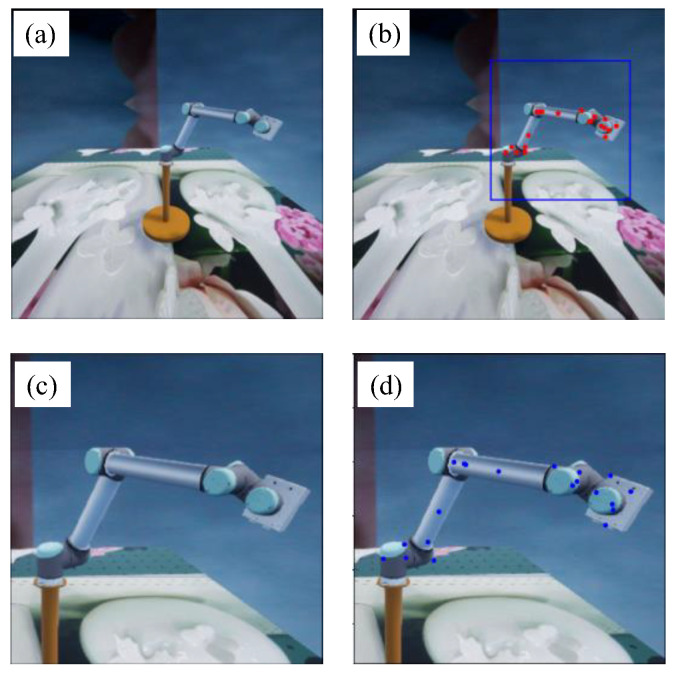
Coarse-to-fine keypoint detection based on dual SRKDNets. (**a**) Original sample image. (**b**) ROI of the robot arm. (**c**) Cropped image of the robot arm ROI. (**d**) Keypoint detection result.

**Figure 7 sensors-24-00305-f007:**
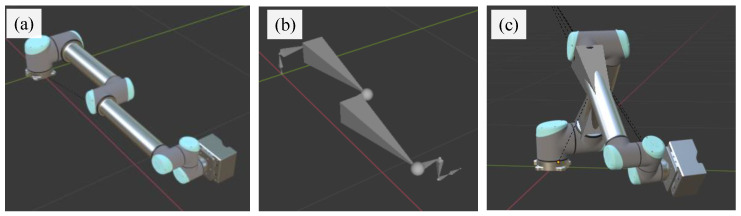
Binding of robot arm and skeleton. (**a**) Robot arm in zero position. (**b**) Movable skeleton of the arm. (**c**) Robot arm moving with the skeleton.

**Figure 8 sensors-24-00305-f008:**
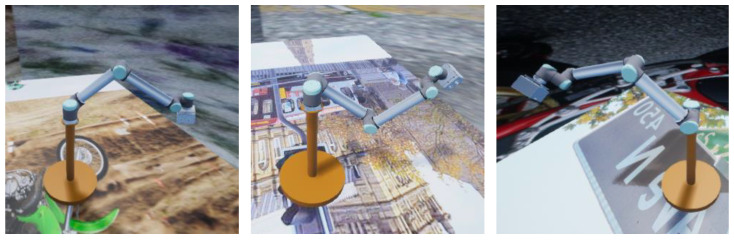
Three synthetic sample images of the robot arm against random backgrounds.

**Figure 9 sensors-24-00305-f009:**
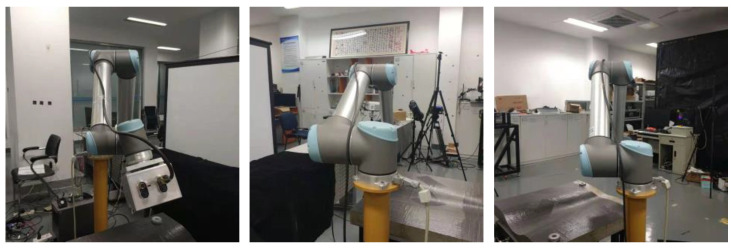
Three examples of robot arm with self-occlusion in real images.

**Figure 10 sensors-24-00305-f010:**
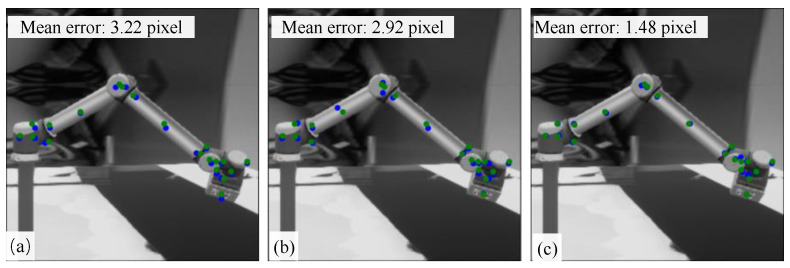
Comparison of keypoint detection results: (**a**) SHNet; (**b**) HRNet; (**c**) full-view SRKDNet.

**Figure 11 sensors-24-00305-f011:**
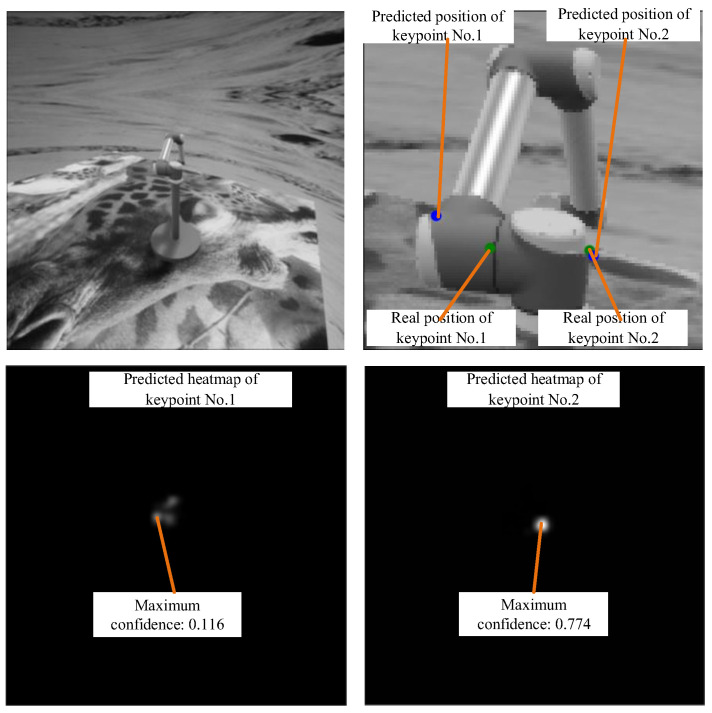
The influence of self-occlusion on keypoint detection.

**Figure 12 sensors-24-00305-f012:**
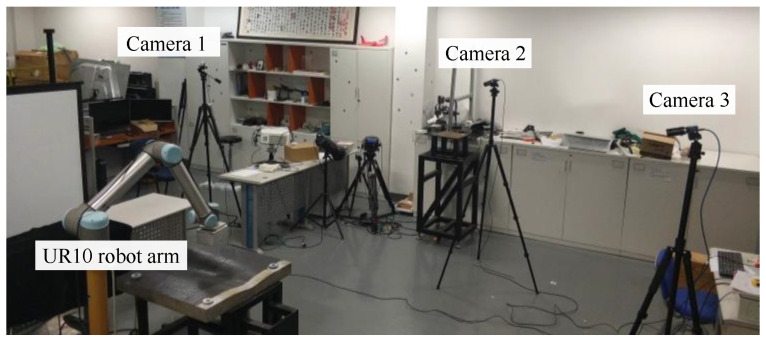
Experiment scene of real robot arm attitude estimation.

**Figure 13 sensors-24-00305-f013:**
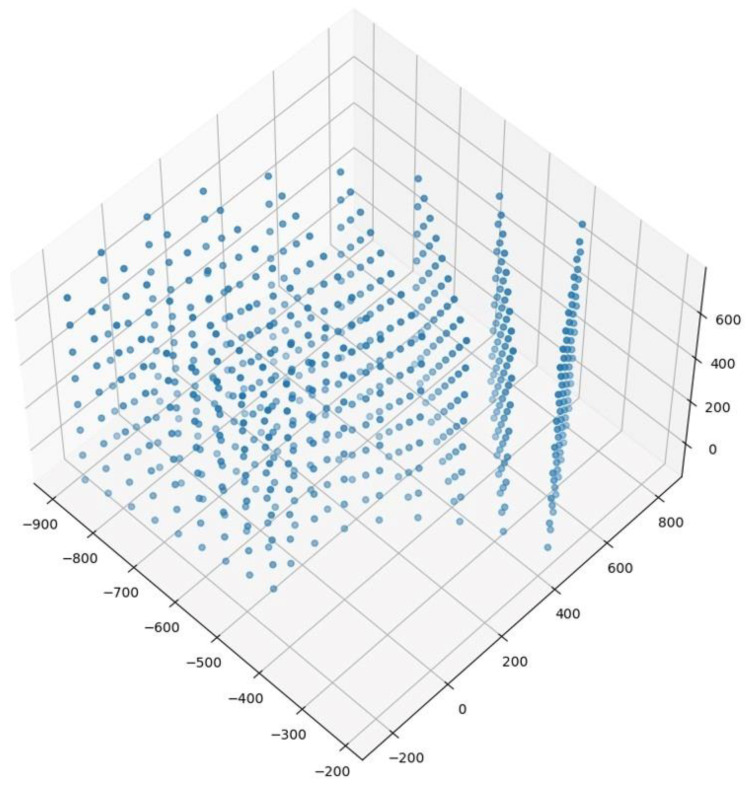
Sample planning of real robot arm in working space.

**Figure 14 sensors-24-00305-f014:**
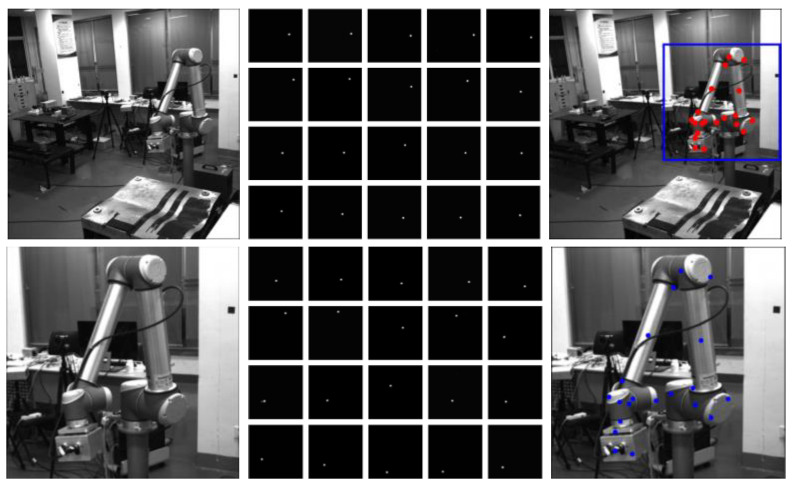
Keypoint detection in realistic scenario.

**Table 1 sensors-24-00305-t001:** Experimental results of keypoint detection on virtual samples.

Methods	PCK@0.2	PCK@0.15	PCK@0.1
SHNet	93.17%	87.27%	66.06%
HRNet	94.91%	88.79%	68.14%
full-view SRKDNet	96.23%	94.33%	89.07%

**Table 2 sensors-24-00305-t002:** Comparison of GPU memory occupation (batch size = 8).

Neural Network Model	GPU Occupation
SHNet (64 × 64)	3397 MB
HRNet (64 × 64)	3031 MB
full-view SRKDNet (256 × 256)	3717 MB

**Table 3 sensors-24-00305-t003:** Comparison of GPU memory occupation (batch size = 2).

Neural Network Model	GPU Occupation
HRNet (256 × 256)	7371 MB
full-view SRKDNet (256 × 256)	1955 MB

**Table 4 sensors-24-00305-t004:** Comparison of full-view SRKDNet only and dual SRKDNets.

Method	PCK@0.15	PCK@0.1	PCK@0.05
full-view SRKDNet	94.33%	89.07%	62.14%
dual SRKDNet	98.69%	97.66%	93.92%

**Table 5 sensors-24-00305-t005:** Robot arm attitude estimation errors based on virtual images (unit: degree).

Method	Joint-1	Joint-2	Joint-3	Joint-4	Joint-5	Joint-6	Average
Single view	0.24	0.42	1.33	5.55	6.72	12.91	4.53
Four views	0.12	0.17	0.30	1.41	1.42	2.95	1.06
Four views + confidence screening	0.05	0.06	0.13	0.70	0.73	1.72	0.57

**Table 6 sensors-24-00305-t006:** Experimental results of keypoint detection for real robot arm.

Method	Virtual Sample Data Used in Training	PCK@0.15	PCK@0.1	PCK@0.05
Full-view SRKDNet	No	92.01%	83.38%	47.27%
Full-view SRKDNet	Yes	95.35%	88.48%	56.26%
Dual SRKDNets	No	91.68%	85.72%	66.50%
Dual SRKDNets	Yes	96.07%	92.51%	78.07%

**Table 7 sensors-24-00305-t007:** Experimental results of real robot arm attitude estimation (unit: degree).

Joint No.	Joint-1	Joint-2	Joint-3	Joint-4	Joint-5	Joint-6	Average
Average error	0.15	0.10	0.15	0.55	0.78	1.47	0.53

**Table 8 sensors-24-00305-t008:** Time consumption of each stage (unit: second).

Stage	Triple-View Keypoint Detection	Robot Arm Attitude Estimation
Time consumption	0.28	0.09

## Data Availability

The data are available from the corresponding author on reasonable request.

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
