# Peer review of "Accurate Robot Arm Attitude Estimation Based on Multi-View Images and Super-Resolution Keypoint Detection Networks"

_sensors, 2024, doi:10.3390/s24010305_

Round 1

Reviewer 1 Report

Comments and Suggestions for Authors

REVIEW OF

Accurate Robot Arm Attitude Estimation Based on Multi-View

Images and Super Resolution Keypoint Detection Networks

BY

Ling Zhou, Ruilin Wang and Liyan Zhang

The article presents the solution of a practical problem using the modern apparatus of convolutional neural networks. The object of observation is a very specific robot manipulator, the purpose of visual observation is to determine its position. To solve the problem, it is proposed to first use a neural network and its architecture is described in detail. By processing the network images coming from surveillance cameras, several estimates of the position are obtained. In the second step, these estimates are combined by a typical Levenberg-Marquardt algorithm.

It should be noted, firstly, the spectacular idea of using software to prepare a training sample for training a neural network. 11,000 marked–up images is a serious groundwork that ensured the success of the study. Without simulation software, it is probably impossible to find the source of such data. The second observation of the reviewer is a very high–quality description and the seriousness of the practical experiment performed.

The article as a whole is well prepared and the reviewer has no fundamental comments. There are two formal suggestions to improve the presentation.

1. Line 117 "... by approximately an order ...". These figures, indeed, show the great advantage of the proposed algorithm. But the authors compare them with universal solutions that were not developed for this task. Well, strictly speaking, there is no advantage ten times. The merit of the authors is not that they "won" the universal algorithm, but that they showed how taking into account the details of the task, the model, and the quality of training improves the result.

2. Like all experts, the authors suffer a little from slang. The reviewer would advise a more critical approach to the presentation and take into account the interests of less prepared readers. For example, a single word network occurs several times. Without specifying that the network is neural, it looks ugly. And it would be even more correct to say right away that convolutional neural networks are understood and used here by machine vision. And follow the terminology without abusing slang.

These suggestions will somewhat improve the presentation, although the article is already very well done.

Reviewer 2 Report

Comments and Suggestions for Authors

The paper explains a two-stage method for UR10 robot arm attitude estimation based on multi-view images, incorporating a Super Resolution Keypoint Detection Network (SRKDNet) and an equation system, with promising results on the robot arm. However, it lacks explicit clarity on its contribution, discussions on generalization and limitations.

1- what are the contributions anf novelty of the proposed method ove SOTA? Define abstract and intorduction.

2- The conribition is not presnted by comparing the recent references in the introduction.

3- The recent references are not 

4- Celarly explain Table 6 in text to understand the reader.

4- What is the future works?

.

Reviewer 3 Report

Comments and Suggestions for Authors

An accurate manipulator Attitude Estimation method is proposed, in which, Multi-View Images and Super Resolution Keypoint Detection Networks are applied.

The authors have conducted many tests and have perfect logic. I think they should provide more explanation on the practical value and novelty of this method. There are many pose estimation methods for manipulators, and it is recommended that the authors provide a comparison.

Comments on the Quality of English Language

  Several grammar errors in the article should be corrected.

Reviewer 4 Report

Comments and Suggestions for Authors

This paper proposes a two-stage robot arm attitude estimation method based on multi-view images. A super-resolution keypoint detection network, which can output high-resolution heatmaps without significantly increasing the computational resource consumption, is proposed to detect the key points of the robot arm in the multi-view images, and an equation system is established to solve the unknown rotation angles of the joints accurately. The main contribution of the paper is how to solve problems such as the selection of the marker points of the robot arm, the internal relationship between the marker points and the kinematic model of the robot arm, the number of samples, the selection method, and the training method (offline or online)? In addition, the paper has the following shortcomings

1.       In the introduction section, "How to accurately estimate the attitude of the robot arms without visual markers is a problem worth studying" (lines 41-42). Is this sentence appropriate? The paper should give the corresponding literature to explain.

2.       What are the criteria for selecting critical points in Section 2.1? Should the connecting rod's center of mass, joint, etc., be considered? Considering the installation of the motor and transmission components, the center of mass of the connecting rod is not necessarily at the center of the connecting rod. In this case, how do you choose? It is suggested to combine the kinematics equation of the manipulator to determine.

3.       In Section 2, the visual calibration method should be given. How to process the data in HRNet, such as determining the number of intermediate layers, weights, training methods, etc.

4.       How do we determine the location of critical points in Figure 6? This problem also appears in Figures 7 and 8.

5.       How do we determine the intrinsic parameter matrix K of the camera? 

6.       Eqs. (2) and (4) use the homogeneous coordinate representation method, but there are errors in dimension; please check and modify it carefully.

7.       Please make a detailed analysis of the experimental results, such as lines 590-594.

Round 2

Reviewer 2 Report

Comments and Suggestions for Authors

The revisions are sufficient.

Reviewer 4 Report

Comments and Suggestions for Authors

All my comments have been properly accommodated.